# Ngn1 inhibits astrogliogenesis through induction of miR-9 during neuronal fate specification

Jing Zhao[1†], Quan Lin[1,2]*[†], Kevin J Kim[1], Faranak D Dardashti[1], Jennifer Kim[1], Fei He[1], Yi Sun[1,2]*

[2]Stem Cell Translational Research Center, Tongji University School of Medicine, Shanghai, China; [1]Department of Psychiatry and Behavioral Sciences and Intellectual Development and Disabilities Research Center, University of California, Los Angeles, Los Angeles, United States

**Abstract** It has been postulated that a proneural factor, neurogenin 1 (Ngn1), simultaneously activates the neurogenic program and inhibits the alternative astrogliogenic program when specifying the neuronal fate. While Ngn1 substantially suppresses the activation of the astrogliogenic Jak-Stat pathway, the underlying molecular mechanism was unknown. Here, by employing in vivo and in vitro approaches, we report that Ngn1 binds to the promoter of a brain-enriched microRNA, miR-9, and activates its expression during neurogenesis. Subsequently, our in vitro study showed that miR-9 directly targets mRNAs of *Lifr-beta*, *Il6st (gp130)*, and *Jak1* to down-regulate these critical upstream components of the Jak-Stat pathway, achieving inhibition of Stat phosphorylation and consequently, suppression of astrogliogenesis. This study revealed Ngn1 modulated non-coding RNA epigenetic regulation during cell fate specifications.

*For correspondence: qlin@ mednet.ucla.edu (QL); ysun@ mednet.ucla.edu (YS)

[†]These authors contributed equally to this work

Competing interests: The authors declare that no competing interests exist.

## Main text

It is known that leukemia inhibitory factor (Lif)-activated Jak-Stat signaling controls the onset of neurogenic-to-astrogliogenic transition (*Bonni et al., 1997*). Lif-activated Jak-Stat signaling pathway in neural stem/progenitor cells (NPCs) starts with three critical components at the cell membrane: the heterodimeric Lif receptor Lifr-beta and Il6st, as well as the receptor-associated Janus kinase (Jak), which are involved in activation of the signaling transducer and activator of transcription (Stat) (*Sun et al., 2001*). We previously demonstrated that the expression of the proneural basic-Helix-Loop-Helix (bHLH) factor neurogenin 1 (Ngn1) inhibits glial differentiation by two distinct mechanisms: (1) Ngn1 sequesters the transcription co-activators Crebbp (CBP)/E1A binding protein p300 (Ep300) away from glial specific promoters; (2) Ngn1 inhibits Stat1/3 phosphorylation (*Sun et al., 2001*). Moreover, we have previously revealed that during astrogliogenic period, phosphor-Stat1/3 directly induces the expression of Il6st and Jak1 to strengthen Stat signaling and trigger astrogliogenesis (*He et al., 2005*). However, the underlying mechanism of how Ngn1 inhibits Stat1/3 activity to block precocious astrocyte differentiation during the neurogenic period is still elusive.

Our previous and current studies showed that protein and mRNA levels of Ngn1 were high during neurogenesis (E12 to E15) but significantly reduced during astrogliogenesis (P0–P4) (*Figure 1A*, lower panel) (*He et al., 2005*). Recently, a group of small non-coding RNA, microRNA (miRNA), has emerged as a novel neuroepigenetic mechanism that can rapidly fine-tune gene expression to regulate developmental timing and cell fate specification (*Lee et al., 1993*; *Wightman et al., 1993*; *Shibata et al., 2008*). While screening the expression pattern of brain specific miRNAs in developing brains and in adult brains by quantitative PCR (qPCR), we found that

**eLife digest** The brain processes information from all over the body through a complex network of cells called neurons. Other brain cells—including star-shaped cells called astrocytes—support this network. Both neurons and astrocytes originate from the same group of stem cells, which first give rise to neurons in a process called neurogenesis before they switch to producing astrocytes.
A protein called neurogenin 1 promotes neurogenesis and suppresses the formation of astrocytes by regulating the activity of particular genes. It does so by binding to a region within the genes called the promoter.

A cell communication system (or 'signaling pathway') known as the Jak-Stat pathway is required for brain stem cells to make astrocytes. Previous research has shown that neurogenin 1 is present at high levels when stem cells start to make neurons, which leads to the inactivation the Jak-Stat pathway. However, when stem cells start to make astrocytes, the levels of neurogenin 1 decrease and the Jak-Stat pathway is activated. This signaling pathway therefore acts as a switch for the transition from neurogenesis to the formation of astrocytes, but it is not clear exactly how it works.

When a gene is active, its DNA sequence is copied to make molecules of ribonucleic acid (RNA). These molecules can be used as templates to assemble proteins—known as messenger RNAs. Alternatively, they may be processed to make another type of RNA called microRNA, which can switch off the activity of particular genes by promoting the destruction of particular messenger RNAs. Zhao et al. studied neurogenesis in the mouse brain and found that neurogenin 1 can directly bind to the promoter of a gene that makes a microRNA called miR-9.

The experiments show that neurogenin 1 increases the activity of this gene so that the amount of miR-9 in brain stem cells increases during neurogenesis. In turn, this microRNA lowers the activity of several critical genes that encode proteins involved in the Jak-Stat pathway. Zhao et al.'s findings reveal that neurogenin 1 promotes neurogenesis and inhibits astrocyte formation by regulating the production of miR-9. The Jak-Stat pathway plays important roles in nerve injury, neural repair, and the immune system, so drugs that target miR-9 may have the potential to be developed into new therapies to treat diseases that affect the nervous system.

expression of miR-9 showed a bell-shaped pattern with their expression reaching maximum level at E16, right before the onset of astrogliogenesis (*Figure 1A*, upper panel). To analyze temporal expression of miR-9 in developing mouse brain, we carried out in situ hybridization (ISH) analysis. Our ISH data showed that miR-9 was extensively expressed in the ventricular zone (VZ) and subventricular zone (SVZ) progenitors in prenatal brains. In neonatal brains (P1–P7), the expression of miR-9 persisted in the SVZ, but at a lower level (*Figure 1B*). Previous studies by *Britz et al. (2006)* and *Ge et al. (2006)* showed that the Ngn1 was highly expressed in VZ/SVZ progenitors during neurogenic period. Taken together, these evidences implied that the expression pattern of miR-9 somewhat correlated with the expression of Ngn1, compatible with a potential regulatory relationship between Ngn1 and miR-9.

By searching upstream sequences of the transcription start sites (TSS) of all three mouse miR-9 genes, we identified a putative Ngn1 binding site containing the E-box element CATATG located 2.5 kb upstream of the miR-9-2 TSS (*Figure 1C*). Subsequently, we confirmed that Ngn1, but not its DNA-binding mutant form, AQ-Ngn1, specifically activated the miR-9-2 promoter-driven luciferase reporter (*Figure 1C*). As expected, when the E-box element was mutated, Ngn1 activation of the miR-9-2 promoter was abolished (*Figure 1C*). To further confirm direct binding of Ngn1 to the putative binding site within the miR-9-2 promoter, we performed chromatin-immunoprecipitation combined with qPCR (ChIP-qPCR) using cultured mouse E11 cortical NPCs expressing T7-tagged Ngn1. The ChIP-qPCR analysis showed that Ngn1 directly associated with the miR-9-2 putative promoter region encompassing the E-box element, but not to the promoter of an irrelevant/control brain enriched miRNA, miR-99b (*Figure 1D*). To determine whether Ngn1 can directly bind to miR-9-2 promoter in vivo, we performed gene-specific ChIP-qPCR experiments using mouse cortex at both the neurogenic stage (E15) when Ngn1 was robustly expressed and the astrogliogenic stage (P3) when Ngn1 expression was diminished. We found that the miR-9-2 promoter was occupied by endogenous Ngn1 in E15 but not in P3 cortex (*Figure 1E*). Finally, we found that overexpression of Ngn1 significantly

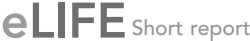

**Figure 1**. Neurogenin 1 (Ngn1) directly regulated miR-9 expression. (**A**) Temporal expression of miR-9 and Ngn1 in the developing and adult mouse cortex. Upper panel: Taqman quantitative PCR (qPCR) analysis of endogenous miR-9 expression levels. Data were normalized to the loading control U6 non-coding RNA. Lower panel: qRT-PCR of endogenous Ngn1 mRNA levels. Data were normalized to loading control Gapdh. Gapdh, glyceraldehyde-3-phosphate dehydrogenase. (**B**) Expression analysis of miR-9 in the prenatal and neonatal mouse cortex by in situ hybridization (ISH). The mature miR-9 was extensively expressed in the VZ/SVZ progenitors in developing cortex. VZ, ventricular zone. SVZ, subventricular zone. Scale bar: 100 μm. (**C**) Ngn1 up-regulated the activity of the miR-9-2 promoter-driven luciferase reporter but not Ngn1 binding site mutant Ngn1 (AQ-Ngn1). (**D**) ChIP-qPCR analysis showed Ngn1 bound to miR-9-2 but not miR-99b promoter region in mouse E11 cortical NPCs overexpressing T7-tagged Ngn1. Data was presented as percentage pull down (IP using T7 antibody) comparing to the input. Negative control IgG pull down was also shown (*p < 0.05, Mann–Whitney test). (**E**) ChIP-qPCR of endogenous Ngn1 associated with miR-9-2 promoter in E15 and in P3 mouse cortexes. Data was presented as percentage pull down (IP

*Figure 1. continued on next page*

*Figure 1. Continued*

using Ngn1 antibody) comparing to the input. Negative control IgG pull down was also shown (*p < 0.05, Mann–Whitney test). (**F**) Taqman qPCR analysis of the expression level of miR-9 in mouse NPCs in the presence of T7-tagged Ngn1 (*p < 0.02, Mann–Whitney test).

up-regulated miR-9 expression (*Figure 1F*). Collectively, these results demonstrated that Ngn1 could directly control the expression of miR-9 during the neurogenic period.

Previous and current studies have showed that (1) Ngn1 inhibited astrogliogenesis partially by inhibiting the astrogliogenic Jak-Stat pathway (*Sun et al., 2001*); (2) miR-9 potentially inhibited Stat activation in mouse embryonic stem cell in vitro (*Krichevsky et al., 2006*); and (3) the expression of miR-9 was controlled by Ngn1 during neurogenesis. Based on these evidences, therefore, we predicted that miR-9 should play essential roles in inhibiting astrocyte differentiation. To test this, we built miR-9 overexpression (miR-9) and miR-9 knockdown sponge (miR-9AS) constructs (*Figure 2—figure supplement 1A*). miR-9, miR-9AS, or control plasmid was delivered into progenitor cells that resided in the ventricular surface by in utero electroporation at E16 (*Figure 2A*). The electroporated cells (GFP+) migrated to layers 2/3 without obvious migration defect in all experimental conditions. We analyzed the ratios of both astrocyte marker+ cells and neuronal marker+ cells vs total transfected cells (GFP+) at P30. Our in vivo study showed that overexpression of miR-9 significantly decreased the number of astrocytes by using three astrocyte markers, Aldh1l1, Slc1a3 (EAAT1), and Gfap, while knockdown of miR-9 increased it (*Figure 2B–D*). We found that neither overexpressing nor knocking down miR-9 had significant effect on the number of Rbfox3+ (NeuN+) neurons (*Figure 2B–D*). By miR-9 ISH combined with Gfap and Rbfox3 immunostaining analysis in adult mouse brains, it became clear that miR-9 was expressed in neurons (Rbfox3+), but not in astrocytes (Gfap+) (*Figure 2—figure supplement 1B*), consistent with the notion that miR-9 inhibited the astroglial program.

To investigate the mechanisms of how miR-9 mediated Ngn1 regulation to prevent precocious astrocyte differentiation, we employed in vitro approaches. miR-9, miR-9AS, or the control plasmid was delivered into E13 cortical progenitor cells by electroporation (*Figure 3A*). The ratios of Gfap+ astrocytes and Map2+ neurons in transfected cells (GFP+) were analyzed at 10 DIV (Days In Vitro). Overexpression of miR-9 significantly decreased the number of astrocytes, whereas knockdown of miR-9 had the opposite effect (*Figure 3A*). Transfection of mouse E11 NPCs with exogenous miR-9 duplexes significantly reduced the number of astrocytes, while having no significant effect on the number of neurons (*Figure 3—figure supplement 1A*). In agreement with our in vivo and in vitro findings, the luciferase reporter assays showed that miR-9 suppressed the activation of glial-specific Gfap promoter, with little effect on the neurogenic promoter Neurod1 (*Figure 3—figure supplement 1B*). While Ngn1 could both activate the neurogenic program and simultaneously inhibit the astrogliogenic program (*Figure 3—figure supplement 1C*), miR-9 appeared to be only involved in the inhibition of glial fate.

We confirmed that transfection of mouse NPCs with exogenous miR-9 duplex blocked phosphorylation of Stat1/3 without altering their protein levels (*Figure 3B*), which is in agreement with the work by *Krichevsky et al. (2006)*. To explore how miR-9 inhibits Stat phosphorylation, we scanned the 3′ UTRs of three upstream components of the Jak-Stat pathway (www.targetscan.com) and found that *Lifr-beta*, *Il6st*, and *Jak1* contain putative miR-9 binding-sites (*Figure 3—figure supplement 2A*). Luciferase reporter assays were used to confirm miR-9 targeting of these components. Transfection of exogenous miR-9 duplex led to an average of 40% decrease in luciferase activities of reporter constructs carrying 3′ UTRs of *Lifr-beta*, *Il6st*, and *Jak1* mRNAs. Such effect was reversed by a miR-9 inhibitor (duplex) (*Figure 3C*). Furthermore, we showed that the effects of miR-9 on the targets were dependent on the miR-9 binding-sites within the 3′ UTRs as the luciferase reporter containing mutant *Lifr-beta* 3′ UTR (with mutated miR-9 binding-site) was resistant to miR-9 inhibition (*Figure 3—figure supplement 2B*). In line with the luciferase assay, we showed that the transfection of mouse NPCs with exogenous miR-9 duplex reduced protein levels of Lifr-beta, Il6st, and Jak1 (*Figure 3D*). We confirmed that Ngn1 also down-regulated the expression levels of Lifr-beta, Il6st, and Jak1 (*Figure 3—figure supplement 2C*) (*Sun et al., 2001*; *He et al., 2005*). We further verified this result by luciferase assays. Overexpression of Ngn1 suppressed the luciferase activities of all three critical component of Jak-Stat pathway (*Figure 3—figure supplement 2D*). In supporting the

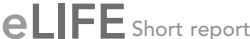

**Figure 2**. miR-9 inhibited astrogliogenesis in vivo. (**A**) Schematic representation of in utero injection of miR-9 constructs followed by electroporation into cortical progenitors resided at the ventricular surface at E16. The right panels showed that cells electroporated at E16 (green) with each condition all migrated to proper cortical layers without overt migration defects. Scale bar: 100 μm. (**B–D**) Left panels showed examples of astrocyte marker+ or neuronal marker+ cells. The arrow pointed Rbfox3+(NeuN) cortical neuron (green and blue). The arrowhead pointed astrocyte marker+ astrocyte (green and red). Scale bar: 10 μm. The arrowhead pointed astrocyte was shown at

*Figure 2. continued on next page*

*Figure 2. Continued*

greater magnification in the last left panel (confocal montage with orthogonal views taken at the center of the cell, scale bar: 5 µm). Right panels showed that overexpression of miR-9 in vivo dramatically attenuated astrogliogenesis, whereas knockdown of miR-9 had the opposite effect (**p < 0.01, *p < 0.5, Mann–Whitney test). Aldh1l1, aldehyde dehydrogenase 1 family, member L1; Slc1a3 (EAAT1), solute carrier family 1 (sodium-dependent glutamate/aspartate transporter 1), member 3; Gfap, glial fibrillary acidic protein.

The following figure supplement is available for figure 2:

**Figure supplement 1**. (**A**) Upper left panels showed construction of miR-9 overexpression plasmid (miR-9) by inserting pre-miR-9-2 and its flanking sequences to a modified lentiviral plasmid FG12.

above-mentioned findings, we showed that co-expression of a constitutively active form of Stat3 (Stat3C) with miR-9 in NPCs could bypass the effect of miR-9 inhibition on astrocyte differentiation (*Figure 3E*). In addition, we previously showed that Jak-Stat signaling has a positive auto-regulation loop, where phosphor-Stat1/3 activates Il6st and Jak1 expression (*He et al., 2005*). Therefore, the DNA binding mutant Ngn1 (AQ-Ngn1), by sequestration of the transcriptional co-activator Crebbp, should also inhibit Stat1/3 mediated activation of Il6st and Jak1, which in turn inhibits Stat1/3 phosphorylation (*Sun et al., 2001*). It appeared that the Ngn1 sequestration regulation and the Ngn1 induced miR-9 regulation acted jointly to suppress Jak-Stat signaling to warrant the neuronal cell fate specification during the time period of neurogenesis (*Figure 3F*).

Since the first miRNA was discovered 20 years ago, miRNA has emerged as a new mechanism of epigenetic regulation that can rapidly respond to extrinsic cues to pattern the activities of particular target protein-coding genes and generate different types of cells over a short period of time (*Lee et al., 1993*; *Wightman et al., 1993*; *Sauvageot and Stiles, 2002*; *Miller and Gauthier, 2007*). This study demonstrated a novel molecular mechanism through which miR-9 mediated the action of Ngn1 by suppressing the expression of three major components of Jak-Stat singling, Lifr-beta, Il6st, and Jak1, which attenuated Stat1/3 phosphorylation to inhibit the astrogliogenic differentiation genes or programs. Our previous and present work revealed that Ngn1 modulated multiple layers of genetic and epigenetic regulations to secure the process of neuronal fate specification.

# Materials and methods

## miRNA isolation and real-time PCR
Small RNA from samples was purified by using the Trizol reagent (Invitrogen, Thermo Fisher Scientific, Waltham, MA). miRNA real-time PCR was performed by using Taqman miRNA assay kit (Applied Biosystems, Thermo Fisher Scientific, Waltham, MA).

## miRNA ISH
LNA 5′-DIG-labeled mercury miR-9 probe (Exiqon, Woburn, MA) were used in ISH on mouse brains. Briefly, brains of CD-1 embryos were dissected and fixed in freshly prepared 4% paraformaldehyde (PFA) (Sigma-Aldrich, St. Louis, MO) for 2 hr at room temperature. The fixed brains were perfused in PBS with 30% sucrose overnight at 4˚C. The next day, the brain tissues were embedded in Sakura Finetek Tissue-Tek O.C.T. Compound and frozen on dry-ice. The frozen brain was sectioned at 10 µm thickness and mounted on Superfrost PLUS slides. After acetylation, the tissue was permeablized by proteinase K at a final concentration of 5 µg/ml and then pre-hybridized by hybridization buffer (50% formamide (Sigma-Aldrich, St. Louis, MO), 5× SSC, 5× Denhardt's solution (Thermo Fisher Scientific, Waltham, MA), 200 µg/ml of yeast RNA (Thermo Fisher Scientific, Waltham, MA), 500 µg/ml of salmon sperm DNA (Thermo Fisher Scientific, Waltham, MA), and 2% Roche blocking reagent (Roche Applied Sciences, Penzberg, Upper Bavaria, Germany)) for 8 hr. For hybridization, 1 pM of the LNA 5′-DIG-labeled mercury probe in 150 µl denaturizing buffer (50% formamide, 5× SSC, 5× Denhardt's solution, 200 µg/ml of yeast RNA, 500 µg/ml of salmon sperm DNA, 2% Roche blocking reagent, 0.25% CHAPS (Sigma-Aldrich, St. Louis, MO), and 0.1% Tween-20 (Sigma-Aldrich, St. Louis, MO)) was added per slide. The hybridization was carried out overnight at 50˚C. After stringency washes, hybridization probe was visualized using anti-DIG-alkaline phosphatase conjugated substrate (Roche Applied Sciences, Penzberg, Upper Bavaria, Germany).

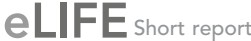

**Figure 3**. miR-9 inhibited astrogliogenesis via targeting three components of Jak-Stat pathway. (**A**) Upper right panels: schematic representation of in vitro delivery of miR-9 constructs into cortical progenitors by electroporation. Lower left panels showed examples of astrocyte marker Gfap+ or neuronal marker Map2+ cells. Right panel showed overexpression of miR-9 significantly reduced the number of astrocytes 10 days after electroporation, whereas knockdown of miR-9 dramatically promoted astrogliogenesis (**p < 0.01, *p < 0.5, Mann–Whitney test). Overexpression of miR-9 in NPCs did not increase the number of neurons. Knockdown of miR-9 reduced the number of Map2+ neurons. Map2, Microtubule-Associated Protein 2. (**B**) Transfection of mouse NPCs with exogenous miR-9 duplex blocked phosphorylation of Stat1/3 without altering their protein levels. (**C**) Luciferase activity of *Lifr-beta*, *Il6st (gp130)*, and *Jak1* 3′ UTR luciferase reporter in the presence of different combinations of control (con), miR-9, miR-9 inhibitor con, and miR-9 inhibitor in mouse NPCs. (**D**) miR-9 inhibited protein levels of Jak-Stat signaling components in NPCs transfected with control or miR-9 duplex. Right panels: western blotting densitometry analysis of protein level changes. Actb (β-actin) serves as the loading control. (**E**) A constitutively active form of Stat3, Stat3C bypassed the effect of miR-9 inhibition on astrocyte differentiation. (**F**) Schematic representation of Ngn1-regulated miR-9 signaling that modulates Stat1/3 phosphorylation to control cell fate specification. Ngn1 up-regulates miR-9 expression during neurogenesis. miR-9 reduces protein levels of Lifr-beta, Il6st, and Jak1 of Jak-Stat signaling pathway by targeting their 3′ UTRs, which in turn abolish Stat1/3 phosphorylation to suppress astrogliogenesis. Crebbp: CREB binding protein, Smad1: Mothers Against DPP Homolog 1 (*Drosophila*), Ep300: E1A Binding Protein p300, E-protein: ubiquitous basic-Helix-Loop-Helix proteins, such as E12 or E47. The dotted arrows show the inhibition regulations.

*Figure 3. Continued*

The following figure supplements are available for figure 3:

**Figure supplement 1**. miR-9 and Ngn1 inhibited astrogliogenesis.

**Figure supplement 2**. miR-9 targeted Jak-Stat signaling pathway.

The immunostaining was carried following the ISH. The primary antibodies used were anti-Gfap polyclonal antibody (Abcam, Cambridge, MA) and anti-Rbfox3 (NeuN) monoclonal antibody (Chemicon, EMD Millipore, Billerica, MA). Fluorophore conjugated secondary antibody were purchased from Invitrogen (Molecular Probe, Thermo Fisher Scientific, Waltham, MA). Images were taken with a Zeiss LSM510 confocal microscope, processed with software Imaris (Bitplane, Switzerland) and composed with adobe Photoshop.

## Plasmid constructions

To construct miR-9 overexpression lentiviral plasmid, the miR-9-2 pre-miRNA fragment with 80 bp flanking sequences was PCR amplified from CD-1 mouse genomic DNA. The PCR product was first cloned into BamH1 and XhoI sites of pBS-hH1 shuttle vector, and then the pre-miR-9 sequence together with human H1 Pol III promoter of the vector pBS-hH1 was digested by Xbal1 and Xho1 and subcloned into FG12 lentiviral vector. miR-9 overexpression forward oligo: CCGGATCCCTGGAGTT CAGCCAGAGGAA. miR-9 overexpression reverse oligo: CCCTCGAGGGTTTTTACTGTCTCTTGGTTGC. To efficiently suppress miR-9 activity, we inserted four bulged miR-9 antisense sequences immediately downstream of the H1 promoter of FG12 (*Ebert et al., 2007*). The control sequence has 3-nucleotide mutations in the miR-9 'seed'-binding region. The bulged miR-9 antisense sequences bind to endogenous mature miR-9, serving as a miR-9 sponge to inhibit miR-9 cellular activity without altering endogenous miR-9 expression. To construct miR-9 knockdown lentiviral plasmid (miR-9AS), the annealed oligonucleotides for miRNA binding sites with 4-nt spacers for bulged sites were cloned into lentiviral vector FG12 (CTL). miR-9AS forward oligo: GATCTCATACAGCTCTTAACCAAAGAATCT-CATACAGCTCTTAACCAAAGAATCTCATACAGCTCTT AACCAAAGAATCTCATACAGCTCTTAACCAAAGATTTTT. miR-9AS reverse oligo: TCGAAAAAATCT TTGGTTAAGAGCTGTATGAGATTCTTTGGTTAAGAGCTGTATGAGATTCTTTGGTTAAGAGCTGTAT GAGATTCTTTGGTTAAGAGCTGTATGA.

To test the efficiency of miR-9 knockdown plasmid, we cloned miR-9 bulged (AS) or miR-9 perfect-matched antisense oligonucleotides duplex into immediate downstream of a firefly luciferase reporter gene of plasmid pIS0 (a gift kindly provided by David Baltimore). miR-9 luciferase assay forward oligo: TCATACAGCTCTTAACCAAAGAATCGTCATACAGCTCTTAACCAAAGAATCGTCATACAGCTAGAT AACCAAAGA; miR-9 luciferase assay reverse oligo: CTAGTCTTTGGTTAAGAGCTGTATGACGATTC TTTGGTTAAGAGCTGTATGACGATTCTTTGGTTAAGAGCTGTATGAAGCT; miR-9 Luciferase assay mutant forward oligo: TCATACAGCTCTTAGCTGAAGAATCGTCATACAGCTCTTAGCTGAAGAATC GTCATACAGCTCTTAGCTGAAGA; miR-9 luciferase assay mutant reverse oligo: TAGTCTTCAGCT AAGAGCTGTATGACGATTCTTCAGCTAAGAGCTGTATGACGATTCTTCAGCTAAGAGCTGTATGAAGCT.

All the lentiviral shuttle vectors and helper vectors were kind gifts from David Baltimore.

## Adenovirus

The T7-tagged mouse Ngn1 was cloned into an adenoviral shuttle vector pMZL6 containing a GFP expression cassette. Control and myc-tagged mouse Ngn1 adenovirus were previously made (*Sun et al., 2001*). Recombinant adenoviruses were made by co-transfection of the shuttle plasmids with the plasmid pBHG10 into HEK293 cells. Viruses were amplified by infecting HEK293 cells and supernatants were harvested, tittered and frozen at −80°C until infection.

## In utero electroporation

For the in utero electroporation, pregnant mice at embryonic day 16 (E16) were deeply anesthetized with isoflurane, uterine horns were carefully exposed through a midline abdominal incision to perform in utero injection and electroporation. Plasmid (2 µl, 2 µg/µl) in saline containing 0.01% fast green was

injected into the lateral ventricle of the embryos through the uterine wall using a NanoFil Syringe with a 36 gouge needle (World Procession Instruments, Sarasota, FL). After injection, electroporation (50 ms square pulses of 40 V with 100 ms, 5 intervals; BTX Electroporator ECM 830 (Harvard Apparatus, Inc., Holliston, MA) was carried out. Then, uterine horns were placed back into the abdominal cavity, and the abdominal wall of the pregnant mouse was sutured. The pups at postnatal day 8 (P8) and day 30 were transcardially fixed with 4% PFA and the brain was continuously fixed in 4% PFA 2 hr at room temperature. Vibratome-sections with GFP positive were subjected to immunohistochemistry.

## Subjects

For in utero electroporation experiments, ICR CD-1 mice were used (Charlies River Laboratories (San Diego, CA). Following the in utero electroporation, the male mice were housed four per cage, maintained on a 12 hr light/dark schedule, and allowed free access to food and water. The protocols are approved by the Institutional Animal Care and Use Committee of the University of California, Los Angeles.

## Cell culture, miRNA mimetics, and electroporation

Mouse (CD1 or Balb/c) cortical neural progenitor cells (NPCs) from E11 cortex were dissected, dissociated, and cultured in DMEM/F12 (Invitrogen) chemically defined medium supplemented with B27. HEK293 and HEK293T cells (ATCC, Manassas, VA) were cultured in DMEM medium with 10% FBS, penicillin/streptomycin and glutamine. LIF (100 ng/ml, R&D Systems, Minneapolis, MN) was used for astrocyte differentiation (1~3-day long-term treatment). Short-term Lif (100 ng/ml) treatment (20 min) was used to detect Stat1/3 phosphorylation. To study the role of miR-9 in regulating astrogliogenesis, CD-1 mouse E13 NPC were isolated and the plasmid (miR-9, miR-9AS, con) was electroporated into the cells using a Nucleofector device (Lonza, Switzerland). To confirm above study, miRNA duplexes and 2′-O-methyl antisense oligonucleotides targeted miR-9 (miR-9 inhibitor) and control (Dharmacon, GE Dharmacon, Lafayette, CO) were electroporated into mouse NPCs by using the nucleofector and mouse neural stem cell nucleofection kit according to the manufacturer's instructions (Lonza). The cells were fixed with 4% PFA for 10 min at room temperature.

## Immunohistochemistry and image processing

The primary antibodies that were used in this study are: anti-Gfap monoclonal antibody (Sigma), anti-Gfap polyclonal antibody (Abcam), anti-Slc1a3 (EAAT1) polyclonal antibody (Abcam), anti-Aldh1l1 polyclonal antibody (Abcam), chicken anti-Map2 antibody (Abcam), anti-Rbfox3 (NeuN) monoclonal antibody (EMD Millipore), and anti-beta III Tubulin (Tuj1) monoclonal antibody (Abcam). For mouse antibody (monoclonal antibody) on mouse tissues, the endogenous mouse IgG blocking method was used to reduce background straining. Briefly, after normal serum blocking and before primary antibody incubation, the mouse brain sections were incubated with unconjugated affiniPure Fab fragment goat Anti-Mouse IgG (H + L) (Jackson ImmunoResearch Labs, West Grove, PA) for 1 hr at room temperature. Fluorophore conjugated secondary antibodies were purchased from Invitrogen (Molecular Probe). Images were taken with a Zeiss LSM510 confocal microscope, processed with software NIH ImageJ and Imaris (Bitplane), and composed with adobe Photoshop.

## Luciferase reporter assays

The 1.9 kb *Gfap* promoter-luciferase reporter construct (pGL3-Gfap) and mouse Neurod1 promoter-luciferase reporter construct (pGL3-Neurod1) were inserted into pGL3 firefly luciferase vector. The mouse miR-9-2 gene promoter sequence and its E-box motif mutant were amplified by PCR and cloned into the pGL3 fly-luciferase construct (Promega, Madison, WI), respectively. The mouse *Il6st (gp130)* 3′ UTR, *Jak1* 3′ UTR, *Lifr-beta* 3′ UTR and miR-9 seed region mutant of *Lifr-beta* 3′ UTR were amplified by PCR and cloned into a firefly luciferase vector, pIS0 vector after luciferase gene (a gift from Dr David Bartel). The miRNA duplex and 2′-O-methyl oligos and fly-luciferase plasmids were co-transfected into E11 mouse NPCs at P2–P4 using Lipofectamine LTX (Invitrogen). TK-pRL Renilla luciferase construct was used as transfection control (Promega). Approximately 24 hr post-transfection, cells were lysed for dual luciferase assays (Promega).

## Western blot analysis

The antibodies used for western were as follows: mouse anti-Gfap (Sigma), rabbit anti-Jak1 (Santa Cruz Biotechnology, Dallas, TX), rabbit anti-Il6st (Santa Cruz, C-20), rabbit anti-Lifr-beta (Santa Cruz, C-19), rabbit anti-Stat1 (BD Biosciences, San Jose, CA), mouse anti-Stat3 (BD Biosciences), mouse anti-phosphotyrosine Stat1 (BD Biosciences), rabbit anti-phosphotyrosine Stat3 (Cell Signaling, Danvers, MA), mouse anti-TuJ1 (Covance, Princeton, NJ) and mouse anti-β-actin (Actb) (Sigma). Secondary goat anti-mouse or anti-rabbit IgG-horseradish antibodies (Calbiochem, EMD Millipore, Billerica, MA) were used, and detection was performed using the ECL plus chemiluminescence (PerkinElmer, Waltham, MA) on X-Omat Blue films (Kodak).

## ChIP

Briefly, ~$10^8$ E11 mouse cortical NPC culture (usually around passage 2) infected with T7-tagged Ngn1 adenoviruses to over 95% infection fate was chemically cross-linked by the additional of 11% formaldehyde solution for 20 min at room temperature. Cells were treated with 1/20 vol of 2.5 M glycine to quench the formaldehyde and washed three times with 1× PBS and pellets were harvested and stored at −80°C prior to use. Cells were re-suspended, lysed, and sonicated to solubilize and shear cross-linked DNA. We used a Branson Sonifier 450 and sonicated at power 5 with a microtip for 7–10 cycles of 30 s pulses (60-s pause between pulses) at 4°C while samples were immersed in an ice bath. After saving 50 μl of whole cell extract (WCE) from each sample to store at −20°C, the rest of sonicated cell lysate was incubated overnight at 4°C with 100 μl Dynal Protein G magnetic beads (Invitrogen) preincubated with 10 μg of the appropriated antibody for at least 8 hr. Beads were then washed five times with RIPA buffer and 1 time with TE containing 50 mM NaCl. Bound complexes were eluted from the beads in elution buffer by heating at 65°C with occasional vortex, and crosslinking was reversed by 6 hr incubation at 65°C. The WCE was also treated for crosslink reversal with additional elution buffer at the same time. Immunoprecipitated DNA and WCE DNA were then purified by treatment with RNaseA, proteinase K and phenol: chloroform: isoamyl alcohol extractions.

## ChIP validations by site-specific qPCR analysis

We used site-specific ChIP-PCR to confirm binding of Ngn1 to miR-9 promoter. Primers were designed to amplify around the E-box location on miR-9-2 promoter. PCR was performed on unamplified DNA samples of several sets of ChIP experiments. The final immunoprecipitated DNA product was dissolved in 70 μl of TE. 2 μl of IP DNA was used in PCR reactions. ~50 ng of the input WCE DNA samples was used. qPCR was performed on iCycler (Bio-Rad) using iQ SYBR Green PCR supermix (Bio-Rad). PCR efficiencies of primers were examined by standard curve of serial-diluted WCEs input and melting curve functionality. The enrichment was calculated as immunoprecipitation signal vs whole cell lysate input (IP/WCE). Antibody used for IP: rabbit anti-Ngn1 (from Greenberg lab, Harvard), rabbit anti-Crebbp (CBP A-22) (Santa Cruz). Normal rabbit anti-IgG antibody (Santa Cruz) was used as the negative. The ChIP-PCR primers for miR-9 were GCCACGGTGCTCTTTAATCT (forward) and TGGTCACAGCATAAACAACTCA (reverse). The ChIP-PCR primers for miR-99b were GGGTCACCCATTTCCTTCTT (forward) and TTCTGAAGGAGGAGGGGATT (reverse).

## Statistical analysis

Wilcoxon-Mann-Whitney test was used to evaluate the statistical significance of differences between groups. Data are presented as mean of fold change compared to control group ±SEM.

## Acknowledgements

This work is supported by Basic Research Program of China (2012CB966303, 2014CB964602) to YES and QL, and EUREKA Grant from NIMH (R01MH084095) and R01 MH066196 to YES. QL is the recipient of 2009 Richard Heyler Award and 2012 Brain & Behavior Research Foundation (NARSAD) Young Investigator Grant. We acknowledge grants from National Natural Science Foundation (31271371, 91319309). We also thank the Intellectual and Developmental Disabilities Research Center (IDDRC center grant: NIH-P30HD004612) at UCLA. We thank Dr. David Bartel (MIT), Dr, David Baltimore (Caltech) and Dr. Caifu Chen (Applied Biosystems) for providing critical reagents for our studies. Last but not least, we would like to thank Donna Crandall of the UCLA-IDDRC Media Core for help with the figures.

## Additional information

### Funding

| Funder | Grant reference | Author |
| --- | --- | --- |
| Brain and Behavior Research Foundation | 19509 | Quan Lin |
| Basic Research Program of China | 2012CB966303, 2014CB964602 | Yi Sun |
| National Institute of Mental Health (NIMH) | R01MH084095 | Yi Sun |

The funders had no role in study design, data collection and interpretation, or the decision to submit the work for publication.

### Author contributions

JZ, Final approval of the version to be published, Acquisition of data, Analysis and interpretation of data, Drafting or revising the article, Contributed unpublished essential data or reagents; QL, Conception and design, Acquisition of data, Analysis and interpretation of data, Drafting or revising the article; KJK, Final approval of the version to be published, Acquisition of data, Drafting or revising the article, Contributed unpublished essential data or reagents; FDD, JK, FH, Final approval of the version to be published, Acquisition of data, Analysis and interpretation of data, Contributed unpublished essential data or reagents; YS, Final approval of the version to be published, Conception and design, Analysis and interpretation of data, Drafting or revising the article

### Ethics

Animal experimentation: This study was performed in strict accordance with the recommendations in the Guide for the Care and Use of Laboratory Animals of the National Institutes of Health. All of the animals were handled according to approved institutional animal care and Chancellor's Animal Research Committee (ARC) protocol #2002001 of UCLA.

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
