## [Decision Letter]

[Editors’ note: this article was originally rejected after discussions between the reviewers, but the authors were invited to resubmit after an appeal against the decision.]

Thank you for choosing to send your work entitled “Ngn1 inhibits astrogliogenesis through induction of miR-9 during neuronal fate specification” for consideration at *eLife*. Your full submission has been evaluated by a Senior Editor, a Reviewing Editor, and three peer reviewers. Based on the reviews below, we regret to inform you that your work will not be considered further for publication in *eLife* at this time. Please note, however, that we would be willing to consider a new submission in the future (with no guarantees of acceptance) should you be able to complete the additional work the reviewers are asking for.

The reviewers indicate potential interest in your study, but they express serious major concerns and do not feel that the work, as it stands, represents a convincing advance over previous work. The reviewers suggest additional experiments to help with this: for example, some are to add more in vivo physiological relevance, some are for more control experiments, and some are to add more stainings with other markers besides GFAP to show that the cells really are astrocytes. We aim to publish articles with a single round of revision that would typically be accomplished within two months, so this means that work that has potential, but in our judgment would need extensive additional work, will not be invited as a resubmission.

Reviewer #1:

This manuscript by Zhao provides data on mechanism by which Ngn1 prevents Stat1/3 phosphorylation, thus inhibiting astrogenesis during a primarily neurogenic stage of development using primarily in vitro approaches. The manuscript extends previous findings that Ngn1 simultaneously promotes neurogenesis and inhibits astrogliogenesis through Stat1/3 phosphorylation (11). It also follows up on another study investigating miR-9 as a brain enriched micro-RNA that prevents Stat3 phosphorylation and could inhibit GFAP expression when cotransfected with miR-124a, but not on its own (Krichevsky, 2009).

This study extends the mechanism of neuron-glial fate choice. The major strength comprises excellent in vitro culture and biochemical data showing that miR-9 inhibits astrogliogenesis and targets some components of the Jak-Stat pathway to prevent expression of GFAP, an astrocyte marker. However, because the majority of the experiments are performed in vitro, the biological relevance of this pathway in the regulation of neuron-glial fate choice in vivo is not clear. The absence of complete in vivo expression analysis and well-defined gain- and loss-of-function experiments are major deficiencies. As this journal does not generally solicit major revisions, the following comments are provided for the benefit of authors:

1) In Figure 2, in utero electroporation of an miR-9 expression construct at E16.5 decreased number of GFAP positive cells from ∼5% to ∼1% at P40. However, no images comparing control and electroporated are shown. The nature of the experiment prevents much quantitation, or a clear understanding of whether miR-9 is necessary for neurogenesis, neuronal maintenance, or has any physiological relevance to neural development.

Regarding the in vivo studies:

2) What is the identity, location, and morphology of the GFAP positive cells that have been electroporated by miR-9? Are they periventricular or cortical? Why was the analysis done so late (P40) since other lineage stages might have been missed?

3) Related to the above point, the expression pattern of MiR-9 in vivo is not clearly demonstrated. Is it expressed in all astrocytes (note that a marker such as Aldh1l1-GFP should be used to capture early stages of radial glia, protoplasmic as well as fibrous astrocytes), or in regionally specific ways or compartments such as white matter, SVZ, cortex, etc.?

4) Electroporation at E16.5 presumably targets the role of Ngn1 during a primarily neurogenic period, when few astrocytes are being produced in vivo. What would the authors propose in terms of the physiological significance of this pathway at this particular time? Would knockdown of mir-9 be expected to increase the number of neurons at the expense of glia? This should be shown.

5) Loss-of-function studies in vivo are needed to demonstrate the biological significance of the findings.

Regarding both in vivo and in vitro studies:

6) Why is only one marker (GFAP) used? Many new markers of astrocyte fate have been identified, including *Glast*, *Glt1*, *Aldh1l1*, *Acsbg1*, among others. Earlier in development, markers of the glial lineage include *Nf1a/b*, *Sox9*, *Fgfr3*, *Id3*, etc. These are useful in characterization of the entire lineage, whereas GFAP is a late marker and not expressed by many cortical astrocytes. Thus, the lineage analysis is insufficient.

Reviewer #2:

Previous work has indicated that in addition to its well-established function of promoting neural specification during development, Ngn1 inhibits astrocytic differentiation by reducing Jak-STAT signaling. In this manuscript, Zhao et al. identify induction of miR-9 as a mechanism linking Ngn1 to reduced Jak-Stat signaling. They identify a region upstream of the miR-9-2 gene that is bound by Ngn1, with overexpression of Ngn1 increasing miR-9 expression. Electoporation of cortical progenitor cells with miR-9 inhibits astrocyte production (as measured by GFAP expression), and transfection of cells with miR-9 reduces expression of LIFRβ, gp130 and JAK1, an effect likely mediated through direct binding to the 3' UTRs of their mRNAs.

Overall, the results of the manuscript are interesting, and the experiments seem to have been well performed. Nevertheless, in our view, the results fail to fully support the authors' claim that Ngn1 mediated induction of miR-9 controls the neuronal/astrocyte cell fate specification decision, and it is questionable whether in its current form the manuscript is a sufficient step forward to warrant publication in *eLife*.

1) If the neuronal/astrocytic cell fate decision is being modulated by miR-9, wouldn't miR-9 overexpression be expected to lead to an increase in neuronal numbers as well as the decrease in astrocytes (as is the case for overexpression of Ngn1)? As it stands this conclusion seems to be overstated.

2) GFAP (which is under direct regulation by STAT signaling) is used as the sole measure of astrocyte specification. Independent markers (e.g., Slc1a3) should be used to determine whether astrocytes are still specified but fail to express GFAP.

3) Relevant to the above point, the fate of the miR-9 transfected cells is unclear. It would seem they do not become neurons, so do they remain uncommitted and continue to proliferate? Do they undergo apoptosis?

4) As the authors note, Ngn1 may also inhibit astrocyte specification, GFAP expression and Jak-Stat signaling via additional miR-9 independent mechanisms. Why not perform an experiment co-electroporating Ngn1 with the antisense miR-9 sponge in order to determine the contribution of miR-9 to Ngn1's role in inhibition of astrocyte generation?

5) Although the ChIP enrichment for Ngn1 at the region upstream of the miR-9-2 gene appears robust, the actual induction of miR-9 by Ngn1 is a little underwhelming (∼1.5 fold, Figure 1).

Reviewer #3:

In this manuscript, the authors describe a role for the microRNA miR-9 in the regulation of Jak-Stat signaling and glial differentiation in the cerebral cortex. They report that the proneural basic helix-loop-helix protein Ngn1 binds to the promoter of miR-9 and induces its expression in mouse neural precursor cells. They also show that overexpression of miR-9 inhibits the expression of the cytokine receptors LIFRβ and gp130 and the protein kinase Jak1. They conclude that Ngn1-induced miR-9 inhibits Jak-Stat signaling and hence suppresses astrogliogenesis.

The authors report an interesting set of results that should advance our understanding of the temporal regulation of neurogenesis and gliogenesis in the mammalian brain. My major concern is the means of inhibiting endogenous miR-9 is through the use of antisense or sponge technology. These methods could produce non-specific effects, which should be discussed. The authors should test whether endogenous levels of LIFRβ, gp130, and Jak1 are specifically upregulated upon inhibition of endogenous miR-9.

Other comments:

1) Is the E-box in the promoter of the miR-9 gene evolutionarily conserved?

2) In ChIP analyses, the authors should show immunoprecipitation efficiency, as percent of input on the y-axis.

3) What is the effect of miR-9 overexpression, antisense, and sponge on neurogenesis and NPC proliferation?

4) What are the consequences of combined miR-9 and Ngn1-AQ on Jak-Stat signaling?

[Editors’ note: what now follows is the decision letter after the authors submitted for further consideration.]

Thank you for resubmitting your work entitled “Ngn1 inhibits astrogliogenesis through induction of miR-9 during neuronal fate specification” for further consideration at *eLife*. Your revised article has been favorably evaluated by a Senior Editor, a Reviewing Editor, and one reviewer. The manuscript has been improved but there are some remaining issues that need to be addressed before acceptance, as outlined below:

The authors have added in vivo data and used IHC markers of astrocytes including Aldh1l1 (Abcam), EAA1, and GFAP. The new loss-of-function data supports conclusions. However, it is unclear which progenitors actually express Ngn1 and MiR-9 to affect a neuron-glial cell fate choice in vivo.

The paper shows in situ hybridization (ISH) of MiR-9 in neurons in adult brain; however, this is not relevant for the proposed developmental functions. MiR-9 expression peaks at E16 by qPCR and so showing expression by ISH remains a major gap and is needed. While the authors cite technical difficulties, these problems seem overstated given that ISH probes are clearly working in adult tissue and E16 ISH is standard procedure in many labs. The requested data would greatly strengthen the paper.

---

## [Author Response]

[Editors’ note: the author responses to the first round of peer review follow.]

In response to the extensive and insightful comments by the reviewers, we made the following revisions:

1) We completed miR-9 in vivo and in vitro loss-of-function studies to fully address the reviewers’ comments (Figure 2 and Figure 3);

2) We completed the quantification analysis of miR-9-regulated astrogliogenesis and neurogenesis in vivo using a neuronal marker, NeuN and two additional astrocyte markers, ALDH1L1 and EAAT1 (Figure 2);

3) We added a new figure to show the specificity of miR-9 knockdown sponge configuration (Figure 2—figure supplement 1);

4) We added a new ISH/immunostaining figure to show the neuronal specificity of miR-9 expression (Figure 2—figure supplement 1);

5) We modified the Ngn1 ChIP-PCR figures to clarify the pull down efficiency (Figure 1).

Reviewer #1:

*[…] This study extends the mechanism of neuron-glial fate choice. The major strength comprises excellent in vitro culture and biochemical data showing that miR-9 inhibits astrogliogenesis and targets some components of the Jak-Stat pathway to prevent expression of GFAP, an astrocyte marker. However, because the majority of the experiments are performed in vitro, the biological relevance of this pathway in the regulation of neuron-glial fate choice in vivo is not clear. The absence of complete in vivo expression analysis and well-defined gain- and loss-of-function experiments are major deficiencies. As this journal does not generally solicit major revisions, the following comments are provided for the benefit of authors*:

In the revised study, we have successfully completed the in vivo gain- and loss-of-function analysis of miR-9-regulated astrogliogenesis by using three astrocyte markers, ALDH1L1, EAAT1 (GLAST, Slc1a3), as well as GFAP (Figure 2). We believe that the in vivo biological relevance of Ngn1-miR-9-Jak/STAT in regulation of neurogenic-to-astrogliogenic transition is well defined now.

*1) In*
Figure 2*, in utero electroporation of an miR-9 expression construct at E16.5 decreased number of GFAP positive cells from ∼5% to ∼1% at P40. However, no images comparing control and electroporated are shown. The nature of the experiment prevents much quantitation, or a clear understanding of whether miR-9 is necessary for neurogenesis, neuronal maintenance, or has any physiological relevance to neural development*.

We found that overexpression of miR-9 in cortical progenitors started from E16 reduced the number of GFAP+ astrocytes to ∼5% of total GFP+ cells, whereas knockdown of miR-9 significantly increased GFAP+ astrocytes to ∼17% of total GFP+ cells (Figure 2). We added three representative panels to show the efficiency of in utero electroporation (Figure 2).

In utero electroporation is a powerful tool that allows genetic manipulation in vivo in order to study a broad range of questions from genes to circuits and behaviors (Fukuchi-Shimogori and Grove, Science, 2001, 294:1071; Bai et al., Nat. Neuroscience, 2003, 18:169; Luo et al., Cell, 2015, 161:1175; Niwa et al., Neuron, 2015, 65:480; Lin et al., unpublished data). Since the genetic liabilities for brain regional- or cell type-dependent behavioral traits are difficult to test in traditional knockout and transgenic animals, in utero electroporation can be used as an alternative tool to study biological questions and, in particular, brain regions. Currently, lacking of astroglio-lineage specific or cell lineage specific miR-9 knockout transgenic mice make lineage tracing difficult. Moreover, miR-9-2 traditional knockout mice die in the early weeks of life, which also hampers astroglio-lineage tracing in vivo (Shibata et al., J. Neuroscience, 2011, 31:3407 and personal communication). Therefore, in the current study, we decided to use in utero electroporation approach to study the role of miR-9 in regulation of astrogliogenesis in vivo.

The major limitation of in utero electroporation is that this pulse gene delivery approach labels a relatively small number of cells in the cortex. Based on our experience, electroporation of the cortical neural progenitors at E16 labeled layers 2/3 cells. We understand that it is impossible to answer all the questions by using in utero gene delivery approach, however, in our opinion, in utero electroporation is a practical and effective tool to analyze the role of miR-9 in regulating cell fate in vivo.

Regarding the in vivo studies:

2) What is the identity, location, and morphology of the GFAP positive cells that have been electroporated by miR-9? Are they periventricular or cortical? Why was the analysis done so late (P40) since other lineage stages might have been missed?

Cortical progenitors electroporated at E16 migrated to layers2/3 (GFP+) (Figure 2). We found very few GFP+ cells remained in the periventricular zone including corpus callosum. Our GFAP immunostaining showed that in the cortex, GFAP+ cells mainly resided in the periventricular region and the superficial layers (layers 2/3 and the marginal zone) of the cortex. We observed that GFAP/GFP double+ cells in general have small and uniform nuclei with fluffy lamellipodium-like peripheral structure. We could not clearly identify the fine processes solely based on the GFP signal.

We analyzed the cell lineage in adult animals because we sought to study the long-lasting effects of Ngn1-driven miR-9 regulation of astrocyte fate.

*3) Related to the above point, the expression pattern of MiR-9 in vivo is not clearly demonstrated. Is it expressed in all astrocytes (note that a marker such as ALDH1L1-GFP should be used to capture early stages of radial glia, protoplasmic as well as fibrous astrocytes), or in regionally specific ways or compartments such as white matter, SVZ, cortex, etc*.*?*

We added a miR-9 in situ hybridization (ISH) combined with an immunostaining figure to show the expression of miR-9 in the cortical region (Figure 2—figure supplement 1). We showed that in the cortex, miR-9 was expressed in ependymal cells and NeuN+ cortical neurons in layers 2-6, but not in the marginal zone and corpus callosum where abundant GFAP+ cells reside. We could not successfully perform miR-9 ISH and immunostaining with other astrocyte markers, such as ALDH1L1 and EAAT1, mainly due to the hash ISH procedures and the nature of the antibodies.

*4) Electroporation at E16.5 presumably targets the role of Ngn1 during a primarily neurogenic period, when few astrocytes are being produced in vivo. What would the authors propose in terms of the physiological significance of this pathway at this particular time? Would knockdown of mir-9 be expected to increase the number of neurons at the expense of glia? This should be shown*.

We chose the time point E16 because we sought to target astrocyte progenitors in the germinal zone of the cortex, which were believed to raise after the peak of neurogenesis (E14-15) (Sauvageot and Stiles, Current Opinion in Neurobiology, 2002, 12:244-249).

We found that overexpression of miR-9 did not significantly increase the number of cortical neurons both in vivo and in vitro, whereas knockdown of miR-9 did reduce it (Figures 2 and 3). We reason that this could be due to some sort of “ceiling” effect of miR-9 expression during the period of neurogenesis in neuronal progenitors. Our ISH data showed that during this period of time, miR-9 was highly expressed in the germinal zone in mouse brains (Lin at al., unpublished data).

*5) Loss-of-function studies in vivo are needed to demonstrate the biological significance of the findings*.

It has been completed both in vivo and in vitro (Figures 2 and 3).

*Regarding both in vivo and in vitro studies*:

*6) Why is only one marker (GFAP) used? Many new markers of astrocyte fate have been identified, including* Glast*,* Glt1*,* Aldh1l1*,* Acsbg1*, among others. Earlier in development, markers of the glial lineage include* Nf1a/b*,* Sox9*,* Fgfr3*,* Id3*, etc. These are useful in characterization of the entire lineage, whereas GFAP is a late marker and not expressed by many cortical astrocytes. Thus, the lineage analysis is insufficient*.

In the revised study, we used two more astrocyte markers, ALDH1L1 and EAAT1 to show the role of miR-9 in regulating astrogliogenesis (Figure 2).

Reviewer #2:

*[…] Overall, the results of the manuscript are interesting, and the experiments seem to have been well performed. Nevertheless, in our view, the results fail to fully support the authors' claim that Ngn1 mediated induction of miR-9 controls the neuronal/astrocyte cell fate specification decision, and it is questionable whether in its current form the manuscript is a sufficient step forward to warrant publication in* eLife*.*

By using in utero electroporation combined with immunostaining, we have completed the in vivo gain- and loss-of-function analysis to demonstrate the role of miR-9 in regulating astrogliogenesis.

Specifically:

*1) If the neuronal/astrocytic cell fate decision is being modulated by miR-9, wouldn't miR-9 overexpression be expected to lead to an increase in neuronal numbers as well as the decrease in astrocytes (as is the case for overexpression of Ngn1)? As it stands this conclusion seems to be overstated*.

Please see response to point 4 from Reviewer #1.

*2) GFAP (which is under direct regulation by STAT signaling) is used as the sole measure of astrocyte specification. Independent markers (e.g., Slc1a3) should be used to determine whether astrocytes are still specified but fail to express GFAP*.

We analyzed the role of miR-9 in regulating astrogliogenesis using two more astrocytes markers, ALDH1L1 and EAAT1 (Figure 2).

3) Relevant to the above point, the fate of the miR-9 transfected cells is unclear. It would seem they do not become neurons, so do they remain uncommitted and continue to proliferate? Do they undergo apoptosis?

Cortical progenitors in utero electroporated at E16 migrated to layers2/3 (GFP+). We did not observe overt migration and cell number differences among miR-9, miR-9AS, and the control. We found there were very few if any GFP+ cells remained in the SVZ at postnatal day 8 (Figure 2), suggesting that neither knockdown nor overexpression of miR-9 affected cell proliferation when in utero electroporation was carried out at E16. Interestingly, we observed that the total number of astrocyte marker and the neuronal marker NeuN double+ cells is about 80% of total GFP+ cells in all experimental conditions (Figure 2). Although the fate of the remaining ∼20% GFP+ cells in the layers 2/3 have yet to be understood, our study clearly demonstrated that perturbation of miR-9 starting at the later stage of embryonic development impaired astrogliogenesis but not neurogenesis.

4) As the authors note, Ngn1 may also inhibit astrocyte specification, GFAP expression and Jak-Stat signaling via additional miR-9 independent mechanisms. Why not perform an experiment co-electroporating Ngn1 with the antisense miR-9 sponge in order to determine the contribution of miR-9 to Ngn1's role in inhibition of astrocyte generation?

We did not perform the suggested assay for two major reasons:

a) Ngn1 inhibits glial differentiation by at least two mechanisms: the sequestration regulation (Sun at al., Cell, 2001) and transcriptional regulation (Sun et al., Cell, 2001; He et al., Nat. Neuroscience, 2005 and the current study). Ngn1-miR-9-Jak/STAT pathway only contributes partially to the Ngn1 signaling that prevents astrogliogenesis.

b) The in utero electroporation experiment only transfects a relative small number of astrocyte lineage cells (∼10% of total transfected cells) in the cortex. Therefore, the rescue effect of miR-9 antisense sponge in Ngn1 overexpressing cells would be difficult to analysis.

In the current study, by miR-9 and Ngn1 qPCR, Ngn1 ChIP-qPCR, and luciferase assays, we demonstrated that Ngn1 directly induced miR-9 expression during the neurogenic period. By miR-9 gain- and loss-of-function studies in vivo and in vitro, we showed that during the time period of neurogenesis, miR-9 mediated Ngn1 in inhibiting astrogliogenesis. By luciferase assay and western blotting analysis, we showed that miR-9 directly targeted LIFRβ, gp130, and Jak1 of Jak-Stat pathway. We also performed the rescue assay using constitutive active form of STAT3, STAT3C in miR-9 overexpressing cells to close the signaling loop between miR-9 and Jak-Stat. In sum, we provided clear and convincing evidences to demonstrate the role of Ngn1-miR-9-Jak/STAT pathway in regulating neurogenic-to-astrogliogenic transition.

*5) Although the ChIP enrichment for Ngn1 at the region upstream of the miR-9-2 gene appears robust, the actual induction of miR-9 by Ngn1 is a little underwhelming (∼1.5 fold,*
Figure 1*)*.

Our miR-9 qPCR analysis and miRNA next generation sequencing data showed that miR-9 was one of the most abundant expressed miRNAs in the cortex during prenatal and neonatal periods. For example, the reads of miR-9 RNAseq was >10-fold greater than that of a neuronal specific miRNA, miR-124 in the cortex in neonatal cortex. Therefore, although there was >=1.5-fold increase (Figure 1), the absolute increase in miR-9 number was massive.

Reviewer #3:

*[…] The authors report an interesting set of results that should advance our understanding of the temporal regulation of neurogenesis and gliogenesis in the mammalian brain. My major concern is the means of inhibiting endogenous miR-9 is through the use of antisense or sponge technology. These methods could produce non-specific effects, which should be discussed. The authors should test whether endogenous levels of LIFRβ, gp130, and Jak1 are specifically upregulated upon inhibition of endogenous miR-9*.

In Figure 2—figure supplement 1, we demonstrated the efficiency and the specificity of the miR-9 knockdown sponge configuration. Our luciferase assay showed that 3-nt mutations in the miR-9 seed region abolished the binding of overexpressed mature miR-9.

*Other comments*:

1) Is the E-box in the promoter of the miR-9 gene evolutionarily conserved?

There are 2 E-box sequences (CATATG) within 5kb upstream of human miR-9-2 pre-miRNA (around -1350bp and -4200bp). There are 3 E-box sequences within 5kb upstream of Rhesus monkey miR-9-2 pre-miRNA (around -1350bp, -1940bp, and -2970bp). The upstream sequence distance of miR-9-2 (homology) between mouse and human is 79.6%. The upstream sequence distance of miR-9-2 (homology) between mouse and Rhesus monkey is 79.6%.

2) In ChIP analyses, the authors should show immunoprecipitation efficiency, as percent of input on the y-axis.

We revised the Ngn1 ChIP-qPCR figures to demonstrate the ChIP assay pull down efficiency (Figure 1). Since Ngn1 is a transcription factor, the pull down efficiency compared to input is lower than abundant proteins such as histones, etc. However, the percentage is within reasonable range as others have reported in the literature. As a negative control, we also included the relative IgG pull down efficiency side by side for a better understanding of the ChIP experiment.

3) What is the effect of miR-9 overexpression, antisense, and sponge on neurogenesis and NPC proliferation?

Please see response to point 4 from Reviewer #1.

4) What are the consequences of combined miR-9 and Ngn1-AQ on Jak-STAT signaling?

We previously reported that AQ-Ngn1, when overexpressed, could partially reduce STAT1/3 phosphorylation (Sun, Nadal-Vicens et al., Cell, 2001). We have previously also published that Jak-Stat signaling has a positive auto-regulation loop, where phosphor-STAT1/3 activates gp130 and jak1 expression (He et al., Nat. Neuroscience, 2005). Therefore, AQ-Ngn1, by sequestration of the transcriptional coactivator CBP, should inhibit STAT1/3 mediated activation of gp130 and Jak1, which in turn inhibit STAT1/3 phosphorylation. In the current study, we revealed a Ngn1-regualted miR-9 signaling pathway that governs the initiation of astrogliogenic cell fate specification during cortical development (Figure 3). It seemed that the Ngn1 sequestration regulation and the Ngn1-miR-9 regulation acted jointly to warrant the neuronal cell fate specification. Thus, we rationalize that the combination of miR-9 and Ngn1-AQ will strongly suppress Jak-Stat signaling and block astrogliogenesis.

[Editors’ note: the author responses to the re-review follow.]

*The authors have added in vivo data and used IHC markers of astrocytes including Aldh1l1 (Abcam), EAA1, and GFAP. The new loss-of-function data supports conclusions. However, it is unclear which progenitors actually express Ngn1 and MiR-9 to affect a neuron-glial cell fate choice in vivo*.

Temporal analysis of miR-9 expression in mouse cortex by in situ hybridization (ISH) showed that the mature miR-9 was extensively expressed in the VZ/SVZ progenitors. The expression levels of miR-9 decreased dramatically in the cortical germinal zone in neonatal brains (P1 and P7) (Figure 1). Studies by Britz et al. and Ge et al. (our lab) showed that in mid-stage cortical development (E14-E15.5), Ngn1 was highly expressed in the ventricular zone (VZ) and the subventricular zone (SVZ) progenitors. It was less abundant near the ventricular surface (Britz et al., Cerebral Cortex, 2006, 16:i138 and Ge et al., PNAS, 2006, 103:1319).

Our ISH data showed that at the mid-stage of cortical development (E16), the expression level of miR-9 was also low or undetectable in a small subset of the ventricular progenitors (Figure 1). It appeared that this small set of cells expanded in P1 and P7 SVZ, suggesting that these cells may contribute to a portion of astrocytes in the cortex. In the current study, by employing an in utero electroporation approach, we pulse-labeled progenitors in the ventricular surface at E16 that produced astrocytes in the superficial neocortical layers. It seemed that this subset of progenitors expressed low or undetectable Ngn1 and miR-9 during the time period of astrogliogenesis. It would be interesting to examine the fates of this progenitor pool in a following study.

*The paper shows in situ hybridization (ISH) of MiR-9 in neurons in adult brain; however, this is not relevant for the proposed developmental functions. MiR-9 expression peaks at E16 by qPCR and so showing expression by ISH remains a major gap and is needed. While the authors cite technical difficulties, these problems seem overstated given that ISH probes are clearly working in adult tissue and E16 ISH is standard procedure in many labs. The requested data would greatly strengthen the paper*.

We added the temporal ISH analysis of miR-9 expression in developing mouse cortex in Figure 1.